# Secular Trends in Height, Body Mass and Mean Menarche Age in Romanian Children and Adolescents, 1936–2016

**DOI:** 10.3390/ijerph18020490

**Published:** 2021-01-09

**Authors:** Raluca-Monica Pop, Arava Tenenboum, Marian Pop

**Affiliations:** 1Department of Endocrinology, Mures County Hospital, 540139 Târgu Mureș, Romania; 2Research Methodology Department, George Emil Palade University of Medicine, Pharmacy, Science and Technology of Târgu Mureș, 38 Gheorghe Marinescu Street, 540139 Târgu Mureș, Romania; 3General Medicine, George Emil Palade University of Medicine, Pharmacy, Science and Technology of Târgu Mureș, 38 Gheorghe Marinescu Street, 540139 Târgu Mureș, Romania; aravate1@gmail.com; 4Informatics and Biostatistics Department, George Emil Palade University of Medicine, Pharmacy, Science and Technology of Târgu Mureș, 38 Gheorghe Marinescu Street, 540139 Târgu Mureș, Romania; marian.pop@umfst.ro

**Keywords:** secular trend, Romania, children, adolescents

## Abstract

Secular trends in anthropometric parameters have been documented in most European countries, but no data is available regarding Romanian. The aim of the study was to calculate secular trend in height, body mass and mean menarche age for Romanian children and adolescents. Methods: A secondary data analysis was performed using ten data sets for urban and eight data sets for rural boys and girls, age 5–15 years, covering 80 years (1936 to 2016). Secular trend in height (cm/decade), body mass(kg/decade) and mean menarche age (years) were calculated. Results: Overall, there was a positive secular trend for height in both genders, which parallels the gross domestic product (GDP)/capita difference, more pronounced in boys, across all age-groups, with a maximum for 15 years-old boys (~3 cm/decade) and 13 years-old girls (~2 cm/decade). Body mass trend was also positive, more accentuated in the rural population. Mean age at menarche was higher in rural compared to urban girls, had a negative trend with the disappearance of the difference in the latest available data set (2013). Conclusion: In summary, an overall positive and ongoing secular trend in height and body mass was documented in Romanian children and adolescents, especially for the pubertal age-range, in concordance to other western countries, but out of phase by approximately 20 years.

## 1. Introduction

The term “secular trend” refers to differences between groups/populations that are explained largely by differences in birth dates [1]. Temporal variations are described for main chronic conditions and reflect the population health change over time [2]. Secular trends in height and body mass have been documented since the 19th century in most European countries. For the same period studies show that mean age at menarche has been falling, but seems to have stabilized around 13 years [3]. The main determinant of the secular increase was considered to be the improvement in living conditions [3,4], but, different socio-economic groups show same trends within a country, while the same socio-economic groups differ significantly across countries [5] Moreover, secular changes have been documented in African countries as well, even with the major differences existing in economic development, disease distribution, public health, nutrition and physical activity [6]. Hence secular trends are considered to be caused by specific factors varying from one population to another, rather than just by the change in socio-economic conditions. Genetic factors must be taken into account, as the secular trend is considered to continue until the “genetic potential” of a population is reached [5]. However, this theory has been challenged, as deoxyribonucleic acid (DNA) sequence change at a population level is a very lengthy process exceeding the time interval used for secular trend analysis and therefore the latter cannot be influenced by genetic factors [7]. The major exception to this statement is the gene pool change caused by migration [1]. Height of a population has long been considered as a proxy for health and nutrition status [8] and secular trends can be justified by improving access to health, including treatment, hygiene and the reduction in infectious diseases’ prevalence. At the same time, the level of education and the income are two important determinants of childhood growth [9,10]. For the evaluation of income and socio-economic status, the purchasing power parities (PPP) can be used, as they try to equalize the purchasing power of different currencies, by eliminating the differences in price levels among countries [11].

Secular trend in childhood growth and adult height has been extensively studied in western countries, but less information is available for the developing or transitional countries.

In regards to body mass, secular changes have generally occurred proportionally to the gains in height. However, in the past three decades, the trend has accelerated and contributed to the current high prevalence of overweight and obesity [12]. For developing countries, the dual nutrition burden (the co-existence of undernutrition and overweight/obesity in the same population) can play a role as well [13,14]. Multiple nutritional and lifestyle factors contribute to changes in anthropometric parameters—diet modifications, with increased availability of unhealthy and cheaper foods, physical activity decline, but community factors as well, like modernization, living conditions [7,15].

Studies of secular changes in anthropometric and pubertal parameters are useful for providing information on nutritional status in early life and updating growth references [16]. At the same time secular trend is a marker of the population’s health over time, provides information on the link between growth and environmental cues and emphasizes inter-generational relationships in body size [3]. Secular changes should be addressed in the context of demographic and epidemiologic transitions as they reflect the population structure and health status and are therefore probably related to secular trends in growth and maturation. The demographic transition has two phases—the premodern characterized by small differences between births and deaths and the modernizing where births continue to increase and deaths decrease. The epidemiologic transition is defined as the shift from high under five mortality to increased mortality from degenerative and chronic conditions. [17].

To our knowledge no published data is available regarding the secular trend in Romania. The aim of the current secondary data analysis was to provide insights into secular trends of growth and maturation in Romanian children and adolescents for the past century.

## 2. Materials and Methods

Study population—Romania is a country located in the Southeast Europe, a former communist republic, with a change of political regime from 1989 and transition towards democracy. Since 2007 it became a member of the European Union. Its population is on a negative trend, with the latest demographic data showing a tendency to reverse the population pyramid and a natural decrease in the population for the past 18 years [18]. The national gross domestic product per capita has been growing from 1173USD Purchasing Power Parities (PPP) [11,19] to 32,297USD PPP in 2019 [20], with an important decline after the change of regime in 1989 and a milder one after the 2008 economic crisis.

### Data Sets

Apart from the synthetic growth charts developed in 2016 [21], there are no growth standards published for the Romanian pediatric population. The following historical data sets were identified through literature review:1936—anthropometric evaluation of 11,761 boys and 10,723 girls 5–15 years of age covering the whole geographical area with measurements from both urban and rural regions [22] with available mean and standard deviation scores (SDS) for height and body mass.1950–1999—National Institute for Public Health anthropometric evaluation—7 cross-sectional evaluations performed every 7 years comprising around 400,000 subjects for each survey, from birth to 18 years of age divided in rural and urban sets, covering the entire geographical area; mean values for height and body mass available; data was obtained for a fee from the Archive of the National Institute for Public Health—missing data due to archive damage and no electronic database available. Apart from the anthropometric evaluation, data on pubertal development was also collected. For the last cross-sectional study (1999) 300,405 subjects were evaluated, representing 4.2% of the entire 0–18 years of age population [23]. All measurements were performed by trained medical personnel using the same methodology.2010—a sample of 1375 boys and 1476 girls, age 6–19 years, from 6 counties, not covering the whole geographical area with measurements performed by 3D scanning [24]2012—regional growth references for school children—3731 subjects, aged 7–19 years from the western part of Romania [25].2013—regional data set from one county in Transylvania, analyzing 1923 children age 6–14 years both from urban and rural schools; data on age of menarche was used in this analysis [26]; the anthropometric parameters were used in the development of the synthetic growth charts [21]2016—The synthetic growth references which included 8407 subjects age 3–18 years of age with measurements performed between 2011–2016, covering the whole geographical area, with combined measurements for urban and rural children. The methodology for developing synthetic growth references is described elsewhere [27]2018—Data from the study on the biomotric potential of school children developed and carried out by the national Institute of Sport Research which included 275,501 school aged children 7–18 years old with full geographical coverage [28].

Variables analyzed: mean height (cm), mean body mass (kg), mean age at menarche (years and months) where available/recorded, age—defined identically in all data sets, according to their methodology. A 6 years old subject is defined like any subject with the age between 5.5 to 6.49 years.

Data sets were included in the final analysis if the required variables were available for each age group and sex. Where available, data was dichotomized as per environment (urban/rural).

The data set from 2012 was excluded as it only covered one county and the reported anthropometric measurements were unlikely to represent a true national mean. Moreover, results were presented as medians and percentiles. The 2010 data set was excluded as it presented results in age-groups which were not suitable for analysis. The 2018 data set was excluded as well as it yielded implausible values for the secular trend.

For urban children, ten data sets were available (1936, 1950, 1957, 1964, 1971, 1978, 185, 1992, 1999 and 2016), while for rural 8 data sets (1936, 1950, 1957, 1964, 1971, 1978, 1985 and 2016). The 1936 and 2016 data sets are considered as an average for both urban and rural areas as their samples included subjects from both geographical locations.

Data analysis—secular trend for the two main anthropometric parameters was calculated and expressed in centimeters/decade and kilograms/decade according to age, sex and environment. Given that that historical data set from 1936 included children aged 5–15, information on this age range was used for all data sets. The same values were used for both urban and rural analysis for the data sets of 1936 and 2016 as they were obtained by pooled analysis from combined urban-rural samples.

For pubertal evaluation, mean age at menarche was available for 5 data sets—1964, 1971, 1978, 1985 and 2013.

Ethical approval—The secondary data analysis was approved by the local ethics committee (decision no. 321/24.09.2019).

## 3. Results

The secular trends for height and body mass is summarized in Figure 1 and Figure 2 for both genders, separate for urban and rural environment.

### 3.1. Height Secular Trends

Overall, there was a positive secular trend for height in both genders, more pronounced in boys, across all age-groups. For urban boys the increase in height was highest in 15-year-olds (3.06 cm/decade) and lowest for 5-year-olds (0.94 cm/decade), with a linear increase between age-groups. There were two negative trends—in 1985 compared to 1978 and in 1999 compared to 1992. For rural boys, the height increased across all age groups, throughout the 80 years analyzed, with the same extremes—5-year-olds with the lowest increase (1.00 cm/decade) and 15-year-olds with the highest (3.75 cm/decade). The only negative trend was in the data set from 1950, for the 5–7 years’ age group. When comparing urban and rural trends, urban boys had a higher increase at the beginning of the study period and appear to slow down to a maximum of 2 cm/decade, while rural boys have a lower amplitude, but do not seem to slow down, averaging 4 cm/decade in the latest data set.

For urban girls, the increase in mean height was lower than in boys for all age groups, with the same two negative trends. The highest increase was observed for 13-year-olds (2.05 cm/decade), while the lowest was for 5-year-olds (0.81 cm/decade). For rural girls, the same trend as for boys was observed, with two negative trends for the 5–7 years’ age group in 1950 and 1978. The increase peaked at 13 years of age, the same as for urban girls. In urban-rural comparison across the whole period, urban girls’ trend seems to slow down, averaging around 1 cm/decade, while in rural girls the trend is still ongoing averaging more than 2 cm/decade.

### 3.2. Body Mass Secular Trends

The secular trend in body mass was positive for both genders, regardless of the environment. For urban boys, the maximum increase was observed in 15-year-olds (2.68 kg/decade), while the lowest was for 5-year-olds (0.26 kg/decade), with a linear increase across all ages. There was only a negative trend in the 1985 to 1978 comparison, with an important increase in the last decade. For rural boys, the same age groups’ extremes were observed, with an important increase across all age groups in the last decade. There was a clear negative trend in the comparison 1950 to 1936.

For urban girls, the maxim increase followed the height trend, with 1.67 kg/decade for 13-year-olds, while the minimum was observed for 5-year-olds (0.23 kg/decade). There was a negative trend for pre-pubertal age-range in 1978, followed by a pubertal negative trend in 1985, with the same pattern repeating in the 1992 and 1999 data sets. The highest positive trend was observed in the first two data sets, with a less pronounced trend in the latest data available (3.16 kg vs. 1.40 kg/decade). For rural girls, the same negative trend as for boys was observed in the 1950–1936 comparison, with a continuous positive trend in all the following data, with the same age group as boys (13 years) contributing the most.

In rural-urban comparison, for both genders, the amplitude of the increase is similar, but more than double in boys compared to girls—a maximum of 3.95 kg/decade in boys vs. 1.64 kg/decade in girls from urban areas and 3.98 kg/decade in rural boys vs. 1.54 kg/decade in girls from the same environment.

### 3.3. Mean Age at Menarche

There is a negative trend in the age at menarche onset both for urban and rural girls. The environmental gap narrowed from 10 months in 1971 to zero in the latest data set available. There was an acceleration in age at menarche in rural girls, with a one-year advancement in the last 45 years (from 13 years and 10 months in 1971 to 12 years and 8 months in 2013), with only 6 months in urban girls (from 13 years and 4 months in 1964 to 12 years and 8 months in 2013), thus explaining the equalization in the latest data set available (Figure 3).

## 4. Discussion

In epidemiology, “secular trend” is defined as the long-term change in morbidity or mortality rates for a given health-related state or event. A synonym used interchangeably is “temporal variation” [29]. In auxology, the term usually refers to changes in anthropometric characteristics and motor fitness over one or several decades [30].

To our knowledge, this is the first analysis of secular trends in anthropometric parameters of Romanian children and adolescents covering almost a century. Previously available studies compared different geographical areas and yielded similar results [31]. Romania follows the same trend in height observed in western countries throughout the 20th century, with a slowing in girls and an overall tendency towards stopping in urban areas.

Since the unification of Romania in 1918, the political structure has gone through constitutional changes due to two major historical events, the Second World War and the fall of the communist regime in 1989. These events divide the Romanian history in four periods which are characterized by major changes in the socio-economic status of the population related to nutrition, health-care access, hygiene and educational level.
The interwar years (1919–1939).The Second World War and first post-war years (1940–1947).The period of the communist regime (1948–1989) that can be further divided into three periods:The post-war and recovery years (up to the end of the 1950s).The economic boom (during 1960s and 1970s).The crisis period of the 1980s.The period that followed the Romanian Revolution of 1989- the post-communism period [32,33]

The current study comprises data from all these periods and the results regarding the anthropometric parameters (height and body mass) trends follow both the historical periods described above and the evolution of gross domestic product per capita in USD PPP [20], thus confirming that growth is a mirror of the conditions of a society [34].

Previous studies have argued that growth in boys is more responsive to changes in the environment [10]. This would mean that during periods of good socio-economic conditions boys grow faster, while during recession their growth is more affected [3]. Our study showed similar results, albeit only for children in urban areas, thus challenging the hypothesis of sexual dimorphism in child growth. On the other hand, the increase in height was more pronounced for male gender across all age groups analyzed which is consistent with Galton’s hypothesis [35], although the exact sex difference percentage might differ from a generation to the next [10]. Although positive secular trends in childhood growth are described for a great number of countries, with some of them reaching a plateau or even a negative trend towards the end of the 20th century, final adult height shows a less pronounced secular trend, with a North to South gradient across Europe [36,37]. Data on Romanian subjects age 18–24 years old—considered as surrogates for adult height—showed a constant height for males and a negative trend for females in the period 1998–2010 [38], with the mention that the number of subjects analyzed was rather small and all subjects were candidates to higher education admission and therefore, it might not reflect the reality across country.

When comparing the Romanian children’s secular trend to other reported analyses [15,16,39,40,41,42], although the secular trend has been described earlier, the increase is similar, leading to the hypothesis that the crisis period of the 80’s and the beginning of the transition years in Romania are characterized by a plateau in child growth, with the trend resuming parallel with the increase of GDP/capita. Several factors could be taken into account: first, quality of life increased in Romania during the 60’s and 70’s with urbanization and modernization at their maximum; second, during the 80’s a sharp decline was described in living conditions (water and electricity access was rationalized) and nutrition (law enforced caloric recommendations for adults, rationalizing access to main nutrients); third, access to healthcare increased by mandatory rural deployment of young doctors [32]. All these factors might have influenced the children and adolescents’ growth and development.

There are three distinct periods reported in the secular trend in childhood: before the age of 2 years (quite small trend reported across studies), from age 2 years to puberty (the highest trend observed) and post-puberty towards final adult height (where the trend is quite similar to the first periods) [43]. The important trend across puberty is due to an increase in tempo, with children of both sexes growing faster and maturing earlier. The opposing trends in adult height and timing of puberty, described as cohort effect [29] have a positive association at individual level, explained by the age at peak height velocity (PHV) trend [44], The data from different countries are consistent in reporting a trend of 1–2 cm/decade at age 5–7 and 2–3 cm/decade at age 10–14 years, with the maximum increase in mid-puberty [45]. Overall, Romania seems to be still experiencing a positive secular trend in children’s growth which leads to the suggestion of conducting a national cross-sectional anthropometric survey in the near future in order to develop specific growth references, since the current available ones might be unfit for the following generations. This is in concordance with the recent published study analyzing the height trajectories and distance form the World Health Organization standards in school-aged children across the globe [46].

Body mass positive trend continues and appears to accelerate in both genders and with a steady increase in the last decades, following the worldwide trend and explaining the increased prevalence of obesity and overweight reported by previous studies [47,48] and the concept of “secular obesity” [45]. Rural children seem to be the main contributors to the weight trend, as the only negative trend was observed during the period covering the Second World War. However, weight analysis should include nutritional assessment, which was not available for the study period, as well as levels of physical activity, as the increase in weight across all-ages parallels with an increase in sedentary behavior and adiposity and decrease in muscle mass [49] with negative secular changes described for cardiorespiratory endurance [50].

Although pubertal assessment was not available for both genders and does not cover the whole period, the results obtained are in close agreement with those from other countries around the world. Mean age at menarche decrease has been documented both for western countries during the first two thirds of the 20th century with a tendency to stabilize [51], or even increase at the beginning of the 21st century and for the rest of the world, where stabilization does not seem to have occurred [6,52,53,54]. At the same time pubertal onset in girls has a negative secular trend as well with geography and ethnicity having the main influence [55,56]. The latest available data from Romania resemble that in the United States of America 40 years before. Again, the rural girls were advancing faster, with no difference observed in the mean age at menarche between environments during the latest data set analysis [26]. This narrowing of the gap could be explained by the catch-up growth and maturation that depends on social class, with the tendency to nullify the height social inequalities [57].The last aspect to be discussed is addressing the secular changes in growth and maturation of the Romanian children and adolescents in the context of demographic and epidemiologic transitions. Romania experienced a modernizing demographic phase with increased birth-rate and decreased mortality in the late 60’s and currently shifted to a premodern phase of the demographic transition [58]; at the same time the epidemiologic transition was observed before the 80’s and after the fall of the communist regime, with under-five mortality declining (but still highest in the European Union) and mortality increasing for chronic conditions [59]. Height increase followed the modernizing demographic transition and age at menarche declined in the second epidemiologic transition. This is in concordance with results from previous studies conducted on smaller communities [17].

### 4.1. Limitations

Some limitations of the current study must be underlined. First, individual data from the included studies was not available and only age-specific average values were used for analysis. This method was previously used and considered suitable for trend analysis over long periods [16,42]. Second, data available for comparison across the whole period covered only the 5–15 years age range which prevented to directly examine all three periods described in childhood secular trends, especially the final adult height [3]. Third, the missing information regarding the specific demographics for each data set, the lack of information with regards to nutritional status and level of physical activity make the interpretation of the results more difficult

### 4.2. Strengths

The main strength of this study is the long-time interval examined, covering 80 years and including data sets from the main historical periods described in the literature. At the same time, although individual data was not accessible, all the samples included were large and had full national coverage. Moreover, all the measurements included in the analysis were standardized and performed by trained personnel, thus eliminating the potential measurement bias. Last, by taking into account the GDP/capita and discussing the results in parallel with the geopolitical and public health aspects, it provides a broader picture on the factors influencing growth and maturation,

## 5. Conclusions

In summary, an overall positive and ongoing secular trend in height and body mass was documented in Romanian children, especially for the pubertal age-range, in concordance to other western countries, but out of phase by approximately 20 years. The trend in height is slowing in female gender and in urban children, but the weight trend seems to accelerate. Socio-economic factors, including GDP/capita seem to play an important role in child growth as well as the epidemiologic and demographic transitions. As future direction, a national cross-sectional anthropometric and nutritional survey might be recommended in order to update the available height references.

## Figures and Tables

**Figure 1 ijerph-18-00490-f001:**
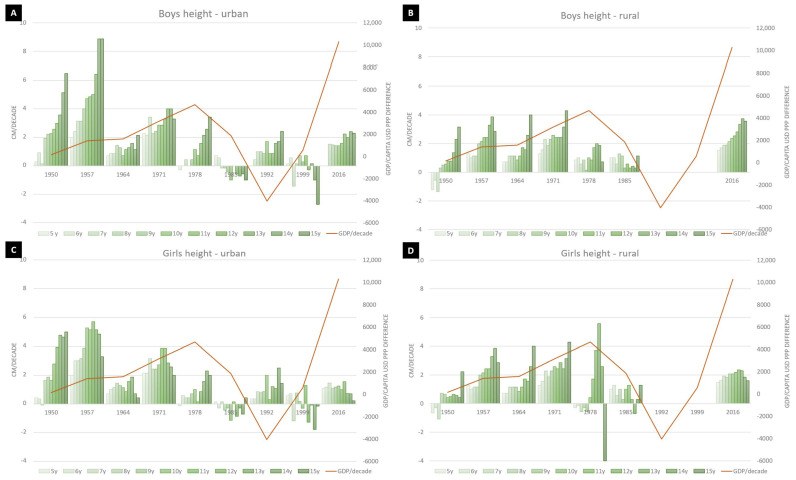
Secular trend for height in (**A**) urban and (**B**) rural boys, respectively (**C**) urban and rural (**D**) girls. The line represents the Gross domestic product GDP/capita in United States Dollars (USD) Purchasing Power Parities (PPP) difference between data points according to [12,13].

**Figure 2 ijerph-18-00490-f002:**
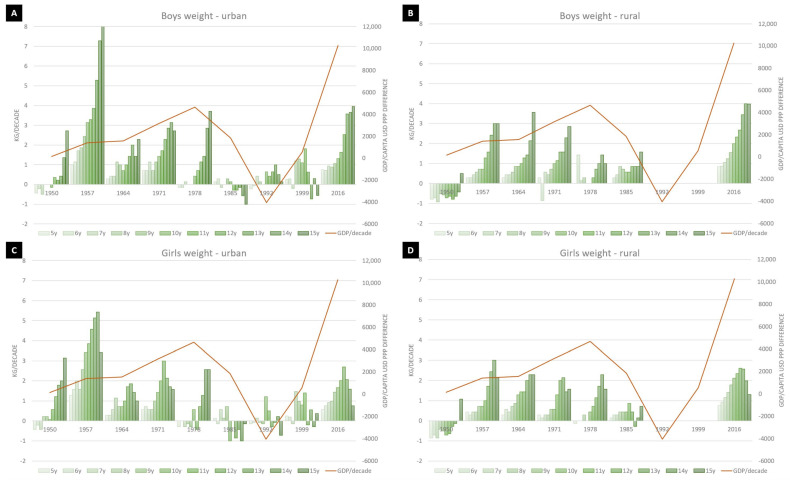
Secular trend for body mass in (**A**) urban and (**B**) rural boys, respectively (**C**) urban and rural (**D**) girls. The line represents the GDP/capita in USD PPP difference between data points according to [12,13].

**Figure 3 ijerph-18-00490-f003:**
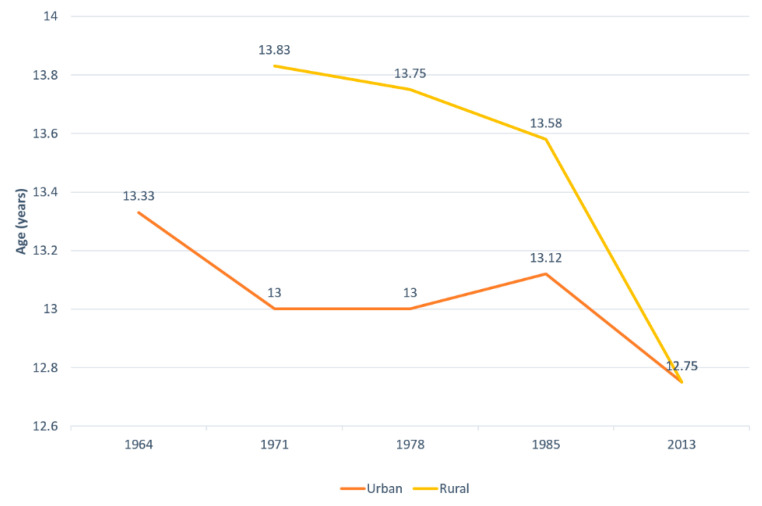
The evolution of mean age at menarche in urban and rural girls.

## Data Availability

Data is contained within the article and all data sources are publicly available.

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
