# Peer review of "Secular Trends in Height, Body Mass and Mean Menarche Age in Romanian Children and Adolescents, 1936–2016"

_ijerph, 2021, doi:10.3390/ijerph18020490_

Round 1
Reviewer 1 Report
A overall good written presentation of the study conducted by the authors.
Only a minor point: figure configurations could be change in order to be more
readable.
Author Response
Response to Reviewer 1 Comments
Only a minor point: figure configurations could be change in order to be more
readable.
Response: We changed the figures' configurations according to suggestion and hopefully have made them more readable. We also rechecked the spelling and made minor adjustments.
Thank you for reviewing our manuscript.
Reviewer 2 Report
This is an observational study examining 'secular trends' in height, weight and age of menarche in a large cohort of Romanian children.
While the data are of interest from public health perspective, it is important to provide a table on baseline features of these children- their overall nutrition and demographics, males vs. females, SES, etc. to make more meaningful interpretation of these data.
The term 'secular trend' and how it relates to nutrition and lifestyle factors must be defined more clearly in the Discussion section.
Author Response
Response to Reviewer 2 Comments
Point 1 While the data are of interest from public health perspective, it is important to provide a table on baseline features of these children- their overall nutrition and demographics, males vs. females, SES, etc. to make more meaningful interpretation of these data.
Response: Thank you for the recommendation. Unfortunately, although very important, data on socio-economic status, nutrition has not been collected in the available data sets. With regards to demographics, the available ones are presented in the data sets description. All data sets came from national representative samples.
Point 2 The term 'secular trend' and how it relates to nutrition and lifestyle factors must be defined more clearly in the Discussion section.
Response: Thank you very much for the suggestion. We updated the discussion section. Lines 213-216 We provided the definition of secular trends in epidemiology and more specific in auxology.
Lines 258-264 and 299-301 We included a discussion of the socio-economic and nutritional factors.
Lines 318-320 – We added the missing information as limitation to our study.
Reviewer 3 Report
Surely, secular trends in various anthropometric parameters have been documented in most European countries and in the world. However, no data regarding Romanian children are available. The authors defined clearly that the aim of the study was to calculate secular trend in height, body mass and mean menarche age for Romanian children. In consequence, the subject of the work is desirable and the findings can aid to define, in the future, strategies to this young population in Romanian.
Title and all the manuscript. Considering the age of the individuals, there is a classification about the groups of persons. World Health Organization. WHO defines 'Adolescents' as individuals in the 10-19 years age group and 'Youth' as the 15-24 year age group. While 'Young People' covers the age range 10-24 years.( https://www.who.int/southeastasia/health-topics/adolescent-health#:~:text=WHO%20defines%20'Adolescents'%20as%20individuals,age%20range%2010%2D24%20years.) Then, about the Title: to clarify about the use of “children”, and I am suggesting to use a classification of the World Health Organization, but the authors can use other classification. I suggest to use “Secular trends in height, weight and mean menarche age in Romanian of Children and Adolescents population, 1936-2016”, but it is only a suggestion to try to clarify the study.
If the suggestion is accepted, to change in all the manuscript.
the Line 18 – to change “weight” to “body mass” here and in all the manuscript.
Line 43- to define “DNA”
Line 47 – to change “explained” to “justified”
Line 51 – to change “between” to “among”
Line 182- The authors must improve the Discussion. At the end of this evaluation, I am suggesting some references to be added,
Line 252 – I suggest to separate “Study limitations and strengths” in “limitations” and “strengths”
Line 279 – I suggest to include the references below. These references might be used in the Introduction and Discussion sections.
1: Eckert-Lind C, Busch AS, Petersen JH, Biro FM, Butler G, Bräuner EV, Juul A. Worldwide Secular Trends in Age at Pubertal Onset Assessed by Breast Development Among Girls: A Systematic Review and Meta-analysis. JAMA Pediatr. 2020 Apr 1;174(4):e195881. doi: 10.1001/jamapediatrics.2019.5881.
2: Mansukoski L, Johnson W. How can two biological variables have opposing secular trends, yet be positively related? A demonstration using timing of puberty and adult height. Ann Hum Biol. 2020 Sep;47(6):549-554. doi: 10.1080/03014460.2020.1795256. Epub 2020 Aug 5. PMID: 32657151.
3: Garenne M. Trends in age at menarche and adult height in selected African countries (1950-1980). Ann Hum Biol. 2020 Feb;47(1):25-31. doi: 10.1080/03014460.2020.1716994.
4: Malina RM, Little BB, Peña Reyes ME. Secular trends are associated with the demographic and epidemiologic transitions in an indigenous community in Oaxaca, Southern Mexico. Am J Phys Anthropol. 2018 Jan;165(1):47-64. doi: 10.1002/ajpa.23326. Epub 2017 Oct 26. PMID: 29072304.
5: Marván ML, Catillo-López RL, Alcalá-Herrera V, Callejo DD. The Decreasing Age at Menarche in Mexico. J Pediatr Adolesc Gynecol. 2016 Oct;29(5):454-457. doi: 10.1016/j.jpag.2016.02.006.
6: Song Y, Ma J, Agardh A, Lau PW, Hu P, Zhang B. Secular trends in age at menarche among Chinese girls from 24 ethnic minorities, 1985 to 2010. Glob Health Action. 2015 Jul 27;8:26929. doi: 10.3402/gha.v8.26929.
7: Fühner T, Kliegl R, Arntz F, Kriemler S, Granacher U. An Update on Secular Trends in Physical Fitness of Children and Adolescents from 1972 to 2015: A Systematic Review. Sports Med. 2020 Nov 7. doi: 10.1007/s40279-020-01373-x. Epub ahead of print. PMID: 33159655.
8: Eckert-Lind C, Busch AS, Petersen JH, Biro FM, Butler G, Bräuner EV, Juul A. Worldwide Secular Trends in Age at Pubertal Onset Assessed by Breast Development Among Girls: A Systematic Review and Meta-analysis. JAMA Pediatr. 2020 Apr 1;174(4):e195881. doi: 10.1001/jamapediatrics.2019.5881.
9: Gondek D, Bann D, Ning K, Grundy E, Ploubidis GB. Post-war (1946-2017) population health change in the United Kingdom: A systematic review. PLoS One. 2019 Jul 3;14(7):e0218991. doi: 10.1371/journal.pone.0218991.
Author Response
Response to Reviewer 3 Comments
Point 1: Title and all the manuscript. Considering the age of the individuals, there is a classification about the groups of persons. World Health Organization. WHO defines 'Adolescents' as individuals in the 10-19 years age group and 'Youth' as the 15-24 year age group. While 'Young People' covers the age range 10-24 years.( https://www.who.int/southeastasia/health-topics/adolescent-health#:~:text=WHO%20defines%20'Adolescents'%20as%20individuals,age%20range%2010%2D24%20years.) Then, about the Title: to clarify about the use of “children”, and I am suggesting to use a classification of the World Health Organization, but the authors can use other classification. I suggest to use “Secular trends in height, weight and mean menarche age in Romanian of Children and Adolescents population, 1936-2016”, but it is only a suggestion to try to clarify the study. If the suggestion is accepted, to change in all the manuscript.
Response: Thank you for the recommendation. We made changes to the title and the rest of the manuscript in accordance.
Point 2 the Line 18 – to change “weight” to “body mass” here and in all the manuscript.
Response: We changed “weight” to “body mass” throughout the manuscript
Point 3 Line 43- to define “DNA”
Response: Line 49 - We defined DNA – deoxyribonucleic acid.
Point 4 Line 47 – to change “explained” to “justified”
Response: Line 53 we changed “explained” to “justified”
Point 5 Line 51 – to change “between” to “among”
Response: Line 58 we changed “between” to “among”
Point 6 Line 182- The authors must improve the Discussion. At the end of this evaluation, I am suggesting some references to be added,
Response: Thank you very much for the suggestion and the list of references. All have been included in the text. Below we provide the exact location in the text for each. We included a paragraph in the introduction discussing body mass trend (lines 64-71) and one describing the demographic and epidemiologic transitions (lines 72-82).
In the discussion section we included the recommended references and discussed their findings (lines213-216, 258-264, 269-271, 286-287, 290-296, 299-301, 302-311, 318-320)
Point 7 Line 252 – I suggest to separate “Study limitations and strengths” in “limitations” and “strengths”
Response: We modified the section and divided in “strengths” and “limitations”
Point 8 Line 279 – I suggest to include the references below. These references might be used in the Introduction and Discussion sections.
Response:
Line 295 Eckert-Lind C, Busch AS, Petersen JH, Biro FM, Butler G, Bräuner EV, Juul A. Worldwide Secular Trends in Age at Pubertal Onset Assessed by Breast Development Among Girls: A Systematic Review and Meta-analysis. JAMA Pediatr. 2020 Apr 1;174(4):e195881. doi: 10.1001/jamapediatrics.2019.5881.
Line 271 Mansukoski L, Johnson W. How can two biological variables have opposing secular trends, yet be positively related? A demonstration using timing of puberty and adult height. Ann Hum Biol. 2020 Sep;47(6):549-554. doi: 10.1080/03014460.2020.1795256. Epub 2020 Aug 5. PMID: 32657151.
Line 45 and 293 Garenne M. Trends in age at menarche and adult height in selected African countries (1950-1980). Ann Hum Biol. 2020 Feb;47(1):25-31. doi: 10.1080/03014460.2020.1716994.
Line 82 and 311 Malina RM, Little BB, Peña Reyes ME. Secular trends are associated with the demographic and epidemiologic transitions in an indigenous community in Oaxaca, Southern Mexico. Am J Phys Anthropol. 2018 Jan;165(1):47-64. doi: 10.1002/ajpa.23326. Epub 2017 Oct 26. PMID: 29072304.
Line 293 Marván ML, Catillo-López RL, Alcalá-Herrera V, Callejo DD. The Decreasing Age at Menarche in Mexico. J Pediatr Adolesc Gynecol. 2016 Oct;29(5):454-457. doi: 10.1016/j.jpag.2016.02.006.
Line 293 Song Y, Ma J, Agardh A, Lau PW, Hu P, Zhang B. Secular trends in age at menarche among Chinese girls from 24 ethnic minorities, 1985 to 2010. Glob Health Action. 2015 Jul 27;8:26929. doi: 10.3402/gha.v8.26929.
Line 287: Fühner T, Kliegl R, Arntz F, Kriemler S, Granacher U. An Update on Secular Trends in Physical Fitness of Children and Adolescents from 1972 to 2015: A Systematic Review. Sports Med. 2020 Nov 7. doi: 10.1007/s40279-020-01373-x. Epub ahead of print. PMID: 33159655.
Line 295 Eckert-Lind C, Busch AS, Petersen JH, Biro FM, Butler G, Bräuner EV, Juul A. Worldwide Secular Trends in Age at Pubertal Onset Assessed by Breast Development Among Girls: A Systematic Review and Meta-analysis. JAMA Pediatr. 2020 Apr 1;174(4):e195881. doi: 10.1001/jamapediatrics.2019.5881.
Line 37 Gondek D, Bann D, Ning K, Grundy E, Ploubidis GB. Post-war (1946-2017) population health change in the United Kingdom: A systematic review. PLoS One. 2019 Jul 3;14(7):e0218991. doi: 10.1371/journal.pone.0218991.
Round 2
Reviewer 2 Report
The authors have addressed the concerns well.